# A high-quality genome assembly highlights the evolutionary history of the great bustard (*Otis tarda*, Otidiformes)

Haoran Luo [1,2,6], Xinrui Jiang[1,6], Boping Li[3,6], Jiahong Wu [1], Jiexin Shen[1], Zaoxu Xu[3], Xiaoping Zhou[2], Minghao Hou[3], Zhen Huang [4,5✉], Xiaobin Ou [3✉] & Luohao Xu [1✉]

Conservation genomics often relies on non-invasive methods to obtain DNA fragments which limit the power of multi-omic analyses for threatened species. Here, we report multi-omic analyses based on a well-preserved great bustard individual (*Otis tarda*, Otidiformes) that was found dead in the mountainous region in Gansu, China. We generate a near-complete genome assembly containing only 18 gaps scattering in 8 out of the 40 assembled chromosomes. We characterize the DNA methylation landscape which is correlated with GC content and gene expression. Our phylogenomic analysis suggests Otidiformes and Musophagiformes are sister groups that diverged from each other 46.3 million years ago. The genetic diversity of great bustard is found the lowest among the four available Otidiformes genomes, possibly due to population declines during past glacial periods. As one of the heaviest migratory birds, great bustard possesses several expanded gene families related to cardiac contraction, actin contraction, calcium ion signaling transduction, as well as positively selected genes enriched for metabolism. Finally, we identify an extremely young evolutionary stratum on the sex chromosome, a rare case among birds. Together, our study provides insights into the conservation genomics, adaption and chromosome evolution of the great bustard.

[1] MOE Key Laboratory of Freshwater Fish Reproduction and Development, Key Laboratory of Aquatic Science of Chongqing, School of Life Sciences, Southwest University, Chongqing 400715, China. [2] Key Laboratory of Ministry of Education for the Coastal and Wetland Ecosystems, College of the Environment and Ecology, Xiamen University, Xiamen 361102, China. [3] Gansu Key Laboratory of Protection and Utilization for Biological Resources and Ecological Restoration, Longdong University, Qingyang, Gansu Province 745000, China. [4] Fujian-Macao Science and Technology Cooperation Base of Traditional Chinese Medicine-Oriented Chronic Disease Prevention and Treatment, Innovation and Transformation Center, Fujian University of Traditional Chinese Medicine, Fuzhou 350108, China. [5] Fujian Key Laboratory of Developmental and Neural Biology, College of Life Sciences, Fujian Normal University, Fuzhou 350117, China. [6]These authors contributed equally: Haoran Luo, Xinrui Jiang, Boping Li. ✉email: zhuang@fjnu.edu.cn; xbou@zju.edu.cn; luohaox@swu.edu.cn

onservation genetics has moved towards an era where high-quality reference genomes are often required[1,2]. For threatened animals, one of major challenges in conservation genomics is to obtain fresh samples for genome sequencing, in particular long-read sequencing[3]. Noninvasive sampling[4], including collecting hairs, feathers[5,6], feces[7], or museum specimens[8] has been widely used in conservation biology, but severe RNA degradation, highly fragmented DNA and heavy contamination limit the performance of high-quality DNA extraction or transcriptome profiling[9,10]. The degeneration of DNA and RNA is much slower in cold temperatures, therefore sampling animals that recently died in frigid zones provides an alternative strategy for obtaining well-preserved DNA or RNA. This has been successful for several mammalian species[11,12], but such effort for avian species has been rare[3].

The recent development of long-read sequencing provides an unprecedented opportunity for complete genome assembly, or telomere-to-telomere genome assembly[13]. This is critical to preserve complete genomic information for endangered species, an endeavor proposed by some initiatives such as the digital Noah's Ark[14]. Nanopore technology provides a fast and high throughput method for sequencing long-reads that can be up to hundreds of kilobases and has been widely used in conservation genomics[15]. In addition, Nanopore reads contain DNA methylation signals, allowing for identification of genome-wide epigenetic modifications[16,17] that are critical for the maintenance of genome stability and gene expression regulation.

Birds have the most streamlined genomes among vertebrates where large-scale genome sequencing projects have been overwhelmingly successful[18,19]. To date, over 500 bird genomes are available, though most were sequenced with short reads[20]. A recent chicken pan-genome study using long-read sequencing, however, suggests bird genomes are far from complete, missing thousands of genes previously thought to be lost[21]. The Vertebrate Genome Project recently also reports massive false gene losses in bird genomes[22]. Moreover, a few microchromosomes have been missing or incomplete in bird genome assemblies likely due to their high GC content and accumulation of simple repeats[22,23].

The great bustard (*Otis tarda*) has been a vulnerable species according to the IUCN Red List of Threatened Species since 1988. The decline of great bustard population in recent years is mainly caused by habitat loss due to human activity[24], collisions with power lines[25] and hunting. Great bustard is amongst the heaviest living flying animals. The male can range in weight from 5.8 to 18 kg[26], and the heaviest verified specimen was about 21 kg, a world record for the heaviest flying bird[27]. The great bustard is also one of the most sexually dimorphic birds in body size, with adult male great bustards being ~2.5 times heavier than females[28]. Despite the heavy body, great bustard is a powered flier and can reach speeds of 48 km/h to 98 km/h during migration[29], and can migrate over 2000 km in northern Mongolia breeding populations[29].

In the January of 2022, we spotted a dead adult male great bustard in a mountainous region of Gansu, China. The cause of death was unidentified, and the date of death was unclear. We immediately relocated the animal to the lab for dissection, and extracted samples for Nanopore ultra-long, Hi-C and RNA-seq library preparation. Fortunately, both DNA and RNA was well preserved, and we were able to retrieve a high-quality genome assembly that contained complete chromosome models. Combining genomic, epigenomic and transcriptomic analyses, we illustrated the genetic diversity, demography, gene expression landscape and the evolutionary history of the sex chromosomes of great bustard.

## Results

**Near-complete genome assembly using ONT-only data.** We extracted high-molecular-weight DNA from thigh muscle tissues of the frozen great bustard (Supplementary Fig. 1), and produced 134.1 Gb (~112X genome coverage, Supplementary Table 1, Supplementary Fig. 2) ONT ultra-long reads. The N50 of the ONT reads reached 37.7 kb, suggesting that long fragments of DNA had been preserved. We used the ONT reads longer than 40 kb (~52X genome coverage) for de novo genome assembly, resulting in a primary assembly with only 129 contigs. The total assembly size is 1.20 Gb, a bit larger than short reads-based estimation (1.09 Gb, Supplementary Fig. 3) by 112.6 Mb. The contig N50 is 41.0 Mb, ranking the fourth longest among more than 500 bird genome assemblies available in NCBI, only next to a chicken[30] and two parrots. To correct potential indel errors, we polished the contigs using 73.3 Gb short-reads generated from the same sample (Supplementary Fig. 4). The BUSCO completeness score is 97.5%, suggesting a high level of genome completeness (Supplementary Table 2).

We further anchored the contigs to chromosome models using 58.6 Gb Hi-C data, generating the final assembly OtiTar_swu. The Hi-C heatmap revealed 40 chromosome models (Fig. 1a, b), consistent with the known karyotype (2n = 80)[31]. Those 40 chromosomes include 39 autosomes and a Z chromosome, accounting for 97.7% of the assembled sequence. The scaffold N50 of OtiTar_swu is 82.8 Mb. Out of the 40 assembled chromosome models, 32 contain zero gaps; the other eight chromosomes have only 18 gaps (Supplementary Fig. 5).

To further evaluate the completeness of chromosomal assembly, we searched for the presence of the telomere repeats (TTAGGG)n and centromeric sequence. We found that the telomeric repeats were present at the ends of 20 chromosomes with a mean length of 2.3 kb (Supplementary Table 3). We identified a putative centromere repeat (Cen191) that was 191 bp long and was present in 38 out of 40 chromosomes. The Cen191 repeats appear at one end in all microchromosomes (Fig. 1a, Supplementary Fig. 6), consistent with the acrocentric morphology of bird microchromosomes[32].

The repetitive sequences occupy 15.1% of the great bustard genome (Supplementary Table 4, in contrast to the mean value of ~9.5% in other birds[19]. This partially explains the larger genome size (~1.20 G) of great bustard compared with the average bird genome size (~1.10 Gb)[33].

**Extremely conserved karyotype throughout avian evolution.** The diploid number of chromosomes (2n = 80) in great bustard is equal to that in emu which is thought to represent the ancestral avian karyotype[34]. In contrast to the complete assembly of chromosomal models in OtiTar_swu, a few small microchromosomes are unfortunately absent in the emu genome assembly. We thus chose to compare OtiTar_swu with a new zebra finch genome assembly (bTaeGut1.4.pri)[22], and a recently published chicken genome that has complete chromosome models[30]. Our synteny analysis showed that all great bustard chromosomes have one-to-one homology with chicken chromosomes except for chr4 and chr4a (Fig. 1c) which were known to have fused in chicken[35,36]. Compared with chicken or great bustard, the zebra finch genome experienced one known fission leading to chr1 and chr1a[35] and one newly identified fusion between two small microchromosomes (Fig. 1c). According to such chromosome comparisons, we inferred that the great bustard likely retains the ancestral avian karyotype, reflecting extreme conservation of karyotype during avian diversification[37,38].

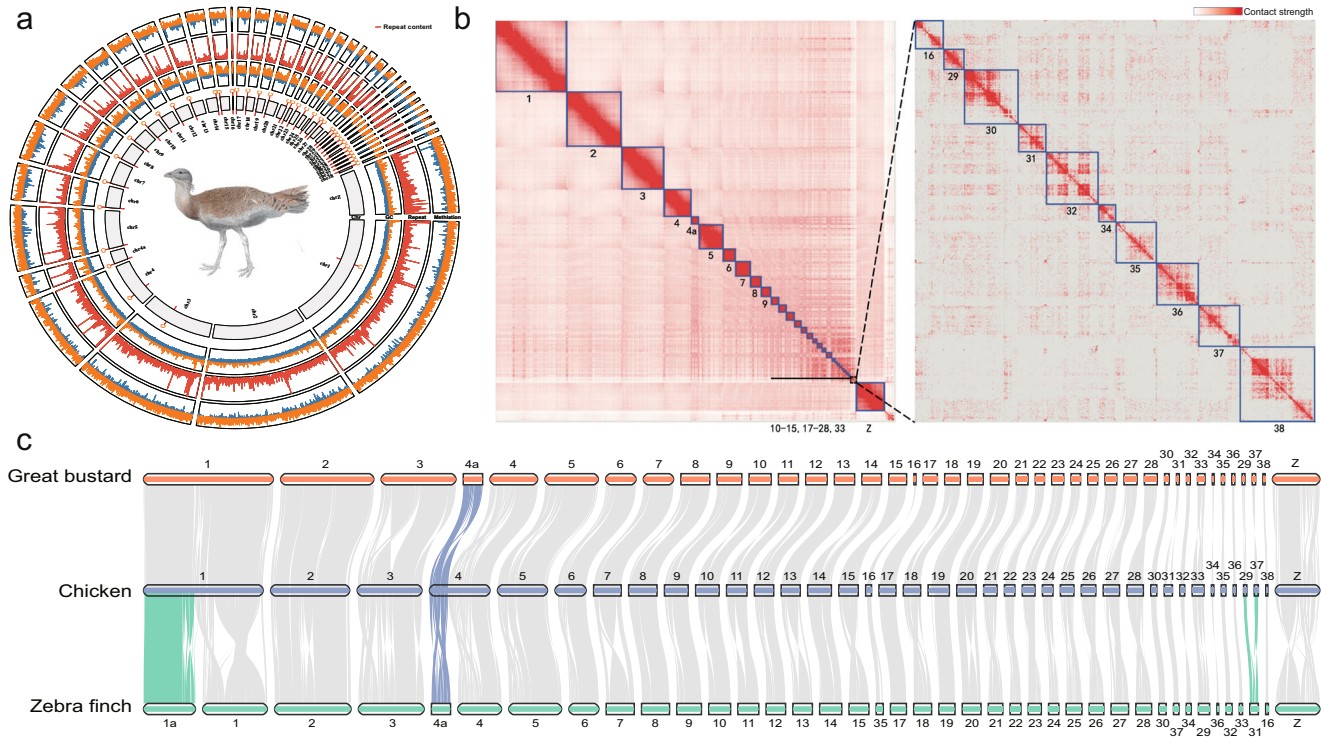

**Fig. 1 A near-complete chromosome-level genome assembly. a** Genomic features of the great bustard visualized by TBtools[126]. Chromosome IDs are labeled at bottom of chromosome bars. In the "Chr" ring, the "C" label represents the centromere position in each chromosome. In the "GC" ring, the orange bars represent regions (in 50 kb windows) with GC-content higher than the genome average (42.9%), while the blue bars represent the below-average regions. In the "Repeat" ring, red bars represent repeat content in 50 Kb window. In the "Methylation" ring, the orange and blue bars represent regions with 5mC higher or lower than 0.5, respectively. **b** Hi-C contact heatmap for the assembled chromosomes. The chromosome IDs are shown under the chromosome models. The right panel shows the zoom-in view for dot chromosomes (Chr16, 29–32, 34–38). **c** Gene synteny blocks among the great bustard, chicken and zebra finch were identified and visualized by MCScan. Horizontal bars represent chromosome models. The synteny blocks highlighted in colors indicate chromosomal fusions.

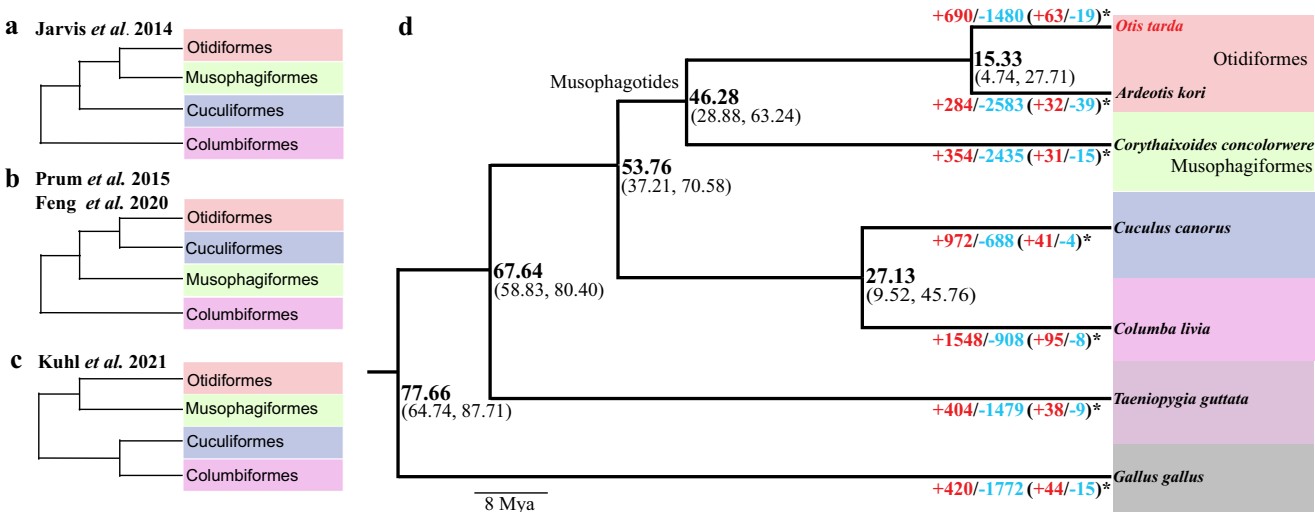

**Fig. 2 Phylogenomic analyses. a–c** Previously proposed phylogenetic relationships for four bird orders. **d** The ultrametric tree reconstructed by this study. Divergence time and confidence interval (95% HPD) are labeled at the nodes. The numbers below branches show the numbers of expanded (blue) and contracted (red) gene families. Numbers of rapid expanded and contracted gene families are labeled with asterisks.

**Phylogenomic analysis resolve the position of Otidiformes.** The phylogenetic position of Otidiformes which great bustard belongs to is one of the unresolved problems in the bird tree of life despite large-scale phylogenomic efforts in the past decade. Using whole-genome alignment data, Jarvis et al. (2014) placed

Musophagiformes as the sister group of Otidiformes[39] (Fig. 2a), while targeted capture data[40] suggested Cuculiformes to be the sister group (Fig. 2b). Though the Musophagotides clade (Musophagiformes+Otidiformes) is further supported by a transcriptome-based phylogeny (Fig. 2c)[41], a more recent

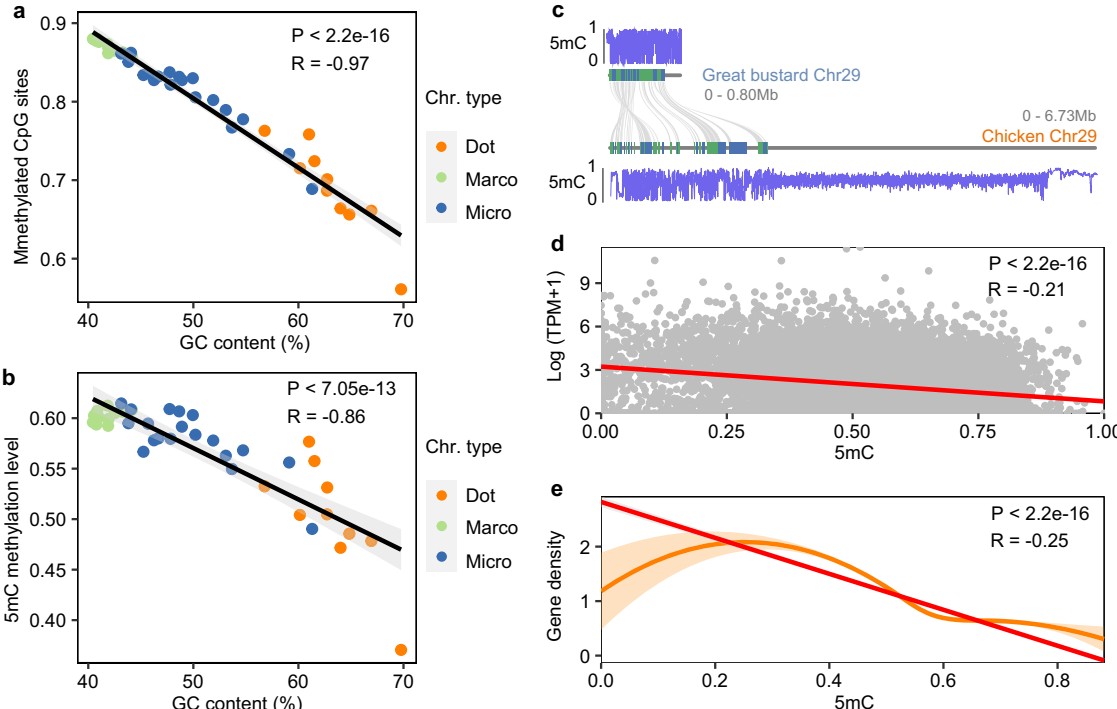

**Fig. 3 Methylation patterns across chromosomes. a** The percentage of methylated CpG sites is negatively correlated with GC content. **b** Chromosome-wide methylation levels show negative correlation with GC content. **c** gene synteny between the chr29 of chicken and great bustard, and the methylation landscapes of the two chromosomes. The chicken chr29, as one example of dot chromosomes, has a large hypermethylated gene-poor region. **d** The expression levels of genes are negatively correlated with the gene-body methylation levels in muscles. **e** The methylation levels are negatively correlated with gene density in 50 kb windows. The shaded areas represent 95% confidence interval in **a**), **b**) and **e**).

phylogenomic study[19] again suggested that Cuculiformes is closer to Otidiformes (Fig. 2b). To resolve such phylogenetic uncertainty, we generated both whole-genome alignments and alignment of coding sequences (CDS) for 5832 single-copy orthologous genes from seven birds. Those birds include two Otidiformes, one Musophagiformes, one Cuculiformes, one Columbiformes species, and two outgroup species (chicken and zebra finch) (Methods). The phylogeny derived from whole-genome alignment and concatenated CDS alignment (Fig. 2a) were in agreement with Jarvis et al. (2014), though concatenated protein sequences-based phylogeny (Fig. 2c) supported the Kulh et al. (2021) phylogeny. Based on the whole-genome or CDS alignment topology we obtained, we estimated the divergence time across the phylogeny. Our analysis showed that Otidiformes diverged from Musophagiformes approximately 46.3 million years ago (Fig. 2d).

**Methylation landscape correlated with GC content and gene expression.** Across the great bustard genome 85.5% of the CpG sites are methylated in the muscle tissue, a percentage somewhat higher than that in human tissues (70–80%)[42]. This percentage varies across chromosomes, with smaller chromosomes having much fewer methylated CpG sites (Fig. 3a). Notably, dot chromosomes (the smallest microchromosomes)[30] have only 68.91% of the CpG sites methylated (Fig. 3a), though they contain a higher density of CpG sites and have higher GC content (Fig. 1b) than microchromosomes or macrochromosomes (Supplementary Tables 5–7). A lower percentage of methylated CpG sites at least in part explains lower chromosome-wide methylation (5mC) levels in dot chromosomes (0.501) than in microchromosomes (0.580) or macrochromosomes (0.602) (Fig. 3b, Supplementary Table 8). This is in contrast to a previous study in chicken where dot chromosomes were found to have higher methylation levels[30].

In chicken dot chromosomes the higher methylation levels are likely driven by the large hypermethylated heterochromatic regions which are, unfortunately, only partially assembled in OtiTar_swu (Fig. 3c). The dot chromosomes in the OtiTar_swu assembly instead contain mostly gene-rich euchromatic sequences (Fig. 3c).

To investigate the relationships between DNA methylation and gene expression[43], we collected RNA-seq data from muscles, the same tissue where we measured methylation. We surprisingly found the RNA still have a moderate quality according to A260/280 values, Q20 and Q30 (Supplementary Table 9) despite the animal exposed in the wild for a while after death. We found that the methylation level and gene expression levels were significantly negatively correlated (Pearson's correlation $P < 2 \times 10^{-16}$, $R = -0.21$) (Fig. 3d), corroborating the role the DNA methylation in repressing gene expression[44]. Our analysis also showed that the methylation level was negatively correlated with gene density (Pearson's correlation $P < 2 \times 10^{-16}$, $R = -0.25$) (Fig. 3e).

**Identifying genes involved in Adaption to powered flight.** Next we sought to identify rapidly expanded gene families and positively selected genes (PSGs) that may play a role in phenotypic diversification and adaptation to the environment[45]. Among the 63 expanded gene families (Supplementary Data 1) are those related to ATPase, short-chain dehydrogenases reductases, mitochondrial inner membrane protein and immunoglobulin. Out of the 24 hierarchy of enriched GO terms for the 432 genes from the expanded gene families, ten directly related to cardiac functions, including regulation of cardiac muscle contraction by calcium ion signaling (GO:0010882) and cardiac muscle contraction (GO:0060048) (Supplementary Fig. 7, Supplementary Data 2).

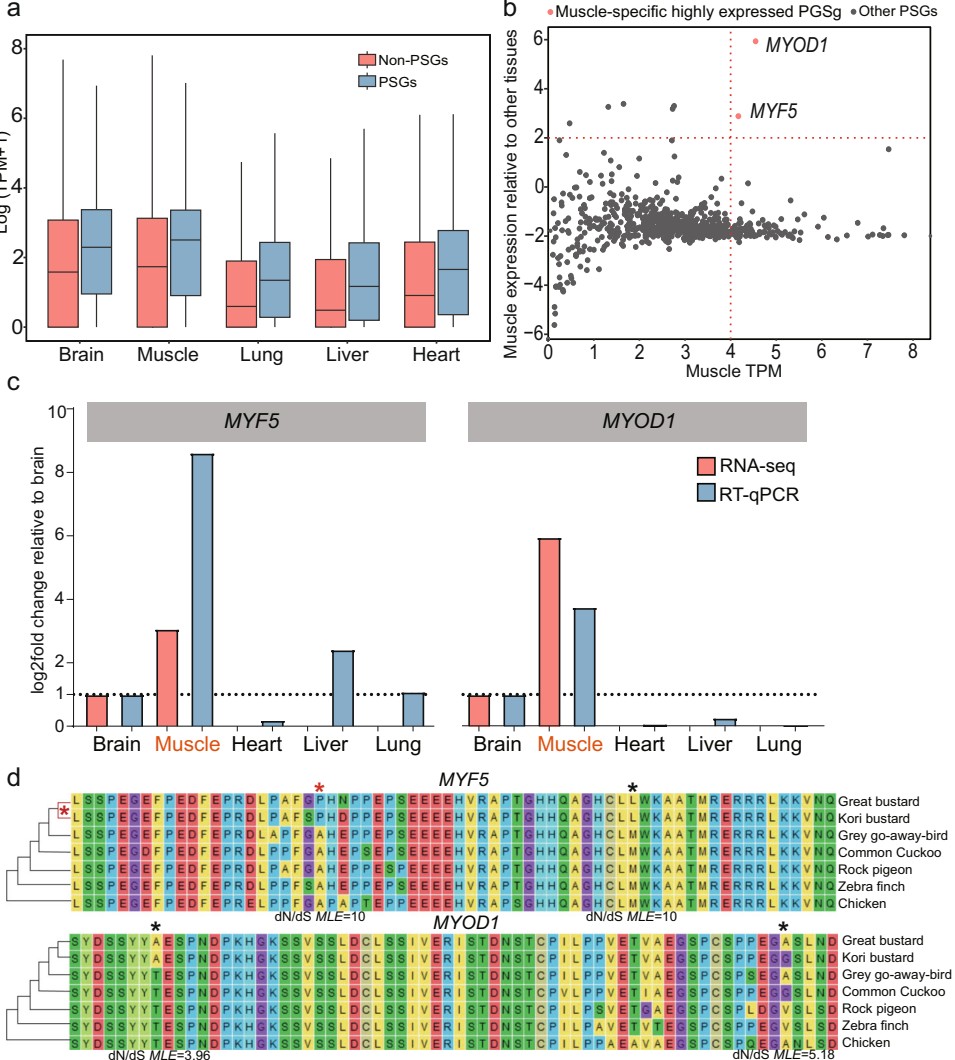

**Fig. 4 Two positively selected genes highly expressed in muscles. a** PSGs have higher expression levels relative to non-PGSs. The bottom and top horizontal lines in the boxplot represent the quartiles, and the middle horizontal lines represent the medians. **b** MYF5 and MYOD1 are specifically highly expressed in muscles, relative to other PSGs. X-axis represents PSGs' expression levels in muscle, and Y-axes represent PSGs' expression levels in muscle divided by the sums of expression levels in other four tissues. Each dot represents one PSG. **c** RT-qPCR verified the trend of expression pattern of MYF5 and MYOD1. **d** Positively selected sites detected by FEL. Sites labeled with asterisks represented positively selected sites.

We detected 763 PSGs after three rounds of robustness tests (Supplementary Data 3), and found that they were enriched for GO terms mainly related to metabolism processes, including protein lipidation (GO:0006497), monocarboxylic acid catabolic process (GO:0072329) and lipid catabolic process (GO:0016042) (Supplementary Fig. 8, Supplementary Data 4). The PSGs generally have higher expression levels than non-PGSs in all five tissues we sampled (FDR < 0.0001, Supplementary Table 10), and have the highest expression levels in muscles (Fig. 4a).

Among the PSGs, two (MYF5 and MYOD1) show specifically high expression in muscles (Fig. 4b), a trend verified by RT-qPCR (Fig. 4c). MYF5 encodes myogenic factor 5, known to be associated with myogenin, involved in positive regulation of muscle cell differentiation and skeletal muscle cell differentiation processes[46,47]. MYOD1 (Myogenic Differentiation 1) promotes muscle-specific target genes transcription[48,49], also plays key role in muscle differentiation[46,47,50]. The use of FEL (Fixed Effects Likelihood) model further suggested that the positively selected sites in MYF5 and MYOD1 were likely selected in the common ancestor of great bustard and kori bustard (Fig. 4d).

**Demographic history and genetic diversity**. The genome-wide nucleotide divergence between great bustard and kori bustard was estimated to be 6.25% based on which we estimated that the neutral mutation rate was $6.11 \times 10^{-9}$ mutations per base per generation in great bustard, and used this value to infer population dynamics. The PSMC analysis suggests that the population effective size ($N_e$) was up to ~80,000 but decreased to ~30,000 from ~250 to ~130 Kya during the RISS glacial period. During the last glacial period (LGP), the $N_e$ stabilized, and gradually recovered to ~40,000. Since then, the $N_e$ fluctuated around ~40,000, similar to the census population size (Fig. 5a). The low resolution of PSMC model in the recent generations, however, warrants future population-based estimation to understand the impact of recent climate changes and human activities on great bustard population dynamics.

Due to the lack of population samples, we used individual genome heterozygosity to approximate the level of genetic diversity[51]. The heterozygosity of great bustard was 0.14%, the lowest among the four Otidiformes genomes we sampled (Fig. 5b). To evaluate whether inbreeding may have caused the

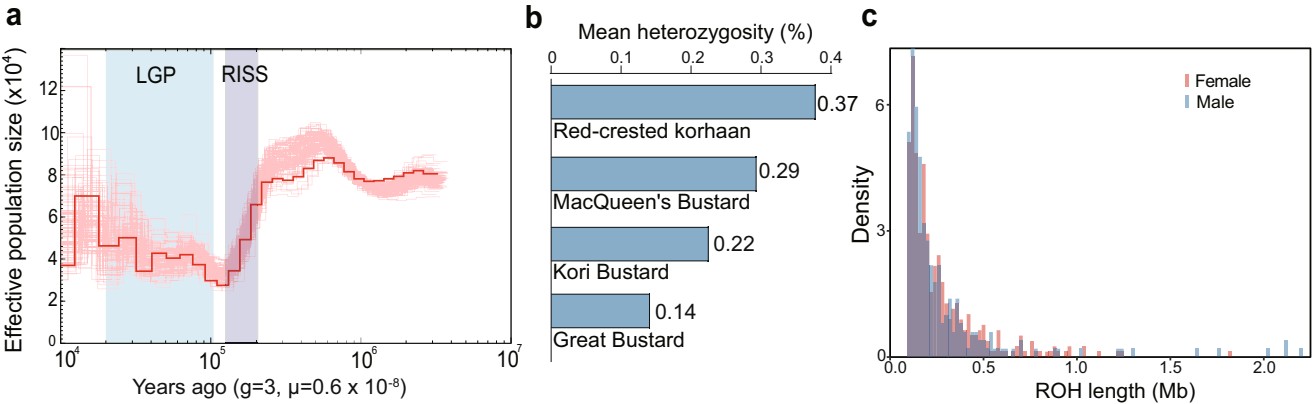

**Fig. 5 Demographic history and heterozygosity. a** The red curve shows the PSMC based estimate for demographic history of the great bustard. The pink curves indicate 100 times of bootstrapping. **b** The great bustard has lowest heterozygosity among four Otidiformes species. **c** ROH length distribution of a male and a female great bustard.

reduced heterozygosity, we calculated the lengths of ROH (run of heterozygosity) using short-read sequencing data from a male (the one used for genome assembly) and a female individual. The average lengths of ROH were 0.27 and 0.36 Mb (Fig. 5c), covering 7.6% and 8.8% of the genome ($F_{ROH}$) for the male and female individual, respectively. The $F_{ROH}$ is smaller than that in kakapo (>15%)[52] or California condor (>20%)[53], but larger than that in Tibetan chicken (4–6%)[54] and many other non-threaten birds. The moderate level of $F_{ROH}$ and relatively short ROH suggested possible inbreeding occurred in distant generations[55], consistent with the ancestral bottleneck inferred from the PSMC analysis.

**A recently evolved sex chromosome stratum.** Neoaves birds share a large non-recombining region of the ZW sex chromosomes, and many lineages further independently experienced an additional event of recombination suppression[56,57], leading to varying lengths of the pseudoautosomal regions (PARs). We demarcated the boundary of PAR in the great bustard by mapping female resequencing data to the male reference genome OtiTar_swu. A ~10 Mb sequence at the edge of the Z chromosome shows a diploid coverage and high female heterozygosity (Fig. 6a, b). The diploid region was further restricted to the first ~2 Mb when a stringent filtering criterion was used by removing the reads with mismatch numbers larger than 2 (Fig. 6c). A likely explanation for the disparity is that the region from ~2 to ~10 Mb (S3) experienced a recent event of recombination suppression between the Z and W, and the W was insufficiently differentiated from the Z, allowing the W-derived reads to be mapped to the Z[58]. In such a case, only the first ~2 Mb should be identified as the PAR.

Similarly, the region from ~10 to ~28 Mb showed a low density of heterozygous sites, in contrast to the rest of the Z being almost hemizygous (Fig. 6b), as well as fluctuation in coverage when a standard mapping criterion was applied (Fig. 6a). This region is likely the third evolutionary stratum (S2) whose range on the Z chromosome matched what has been identified on other Neoaves birds[59]. The boundary of S2 aligned with the breakpoint of an inversion between the Z chromosomes of great bustard and emu (Supplementary Fig. 9) that has been predicted to create the Neognath S1[56].

## Discussion

In this study, we generated a near-complete chromosome-level genome assembly of the great bustard using DNA extracted from

a dead individual found in the wild. The OtiTar_swu is among the most continuous bird genome assemblies, containing all 40 chromosome models. This has been rare in bird genome assemblies because the small microchromosomes, or dot chromosomes, are difficult to be assembled due to their high GC content and abundant tandem repeats[22,23,30]. We relied on ONT ultra-long sequencing data alone for generating such a continuous genome assembly. Though ONT reads suffer from lower read accuracy compared with HiFi reads, they may have helped resolve the assembly of dot chromosomes[30]. The ONT data alone has sufficient power to resolve centromeric and telomeric sequences for some of the chromosomes of great bustard, and the use of R10.4 flowcell is expected to further increase the capability of ONT data to resolve complex regions[60].

Great bustard has received increasing attention in recent years, and large efforts have been made in species conservation, including re-introduction (according to the Royal Society for the Protection of Birds). Our study provides a high-quality genome assembly of great bustard which will serve as an important genomic resource for conservation management. The low genetic diversity of great bustard revealed by our analysis is alarming, demanding future surveys of genetic diversity among great bustard populations.

The genomic sequence also allows for a comparative genomic analysis to search for clues of genetic adaptation to powered flight in great bustard that is among the heaviest flying birds. We identified *MYF5* and *MYOD1* that were positively selected in great bustard and have high expression levels in muscles. Both genes are essential for muscle cell fate specification in bird embryos[61,62]. Furthermore, we identified several expanded gene families that were related to cardiac muscle contraction. Those analyses suggest enhanced muscle function likely plays a role in great bustard long-range powered flight, and future in-depth functional analysis is needed to illustrate to molecular mechanisms of *MYF5* and *MYOD1* adaptation.

Our comparative genomic analyses have also provided insights into avian genome evolution. Because the sampled individual is male, the assembled genome lacks a female-limited W chromosome. Nevertheless, we were able to infer the evolutionary strata on the Z chromosome based on sequencing coverage and heterozygosity. Further efforts are needed to assemble a female genome in order to analyze the gene content and evolution of the W chromosome[63], and similar analyses are required for closely related lineages, such as Musophagiformes, to illustrate a more complete evolutionary history of sex chromosomes. Our chromosomal comparison across Neognaths also provide direct

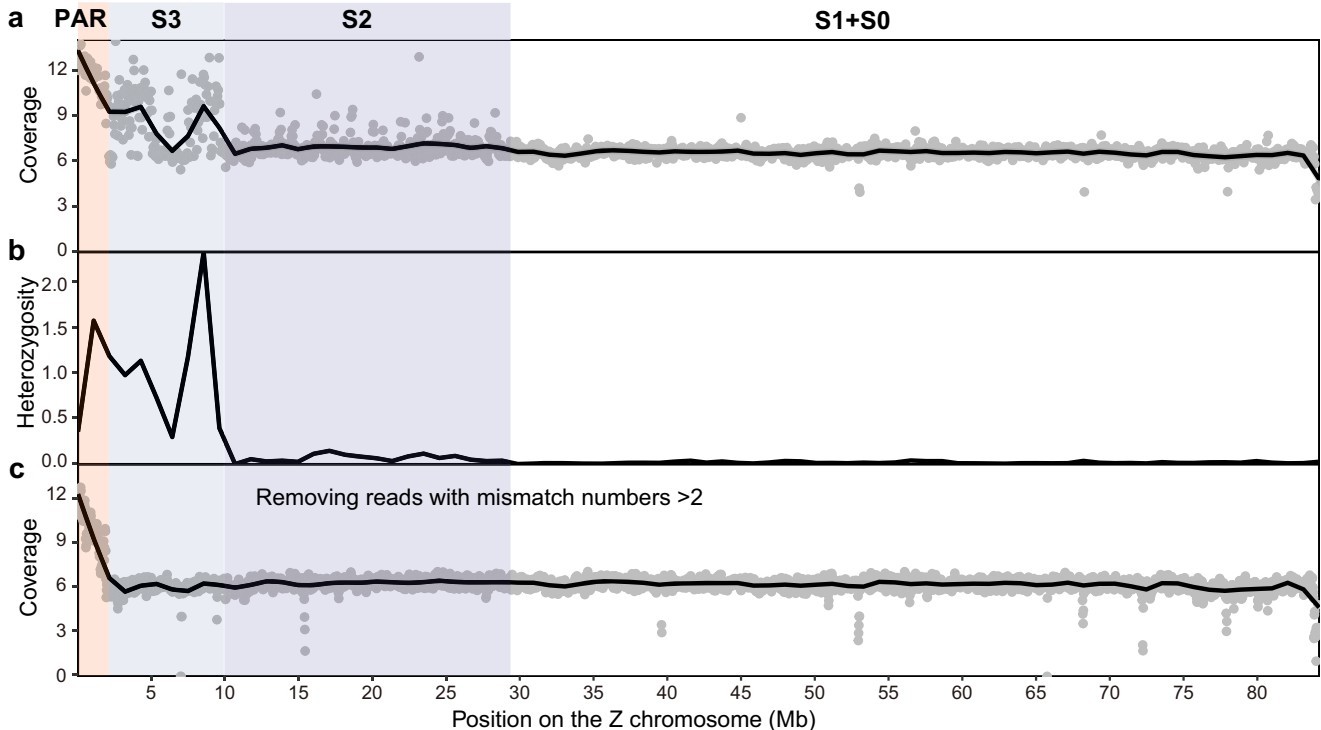

**Fig. 6 Identification of a young evolutionary stratum on the sex chromosome. a** The coverage of female re-sequencing data calculated in 50 kb windows. **b** female heterozygosity calculated in 50 kb windows on the Z chromosome. **c** similar to **a**) but the reads with mismatch number larger than 2 were removed from the alignments before calculating coverage.

genomic evidence for the evolutionary stasis of avian chromosome[37], including the smallest microchromosomes that receive little study until recently[30].

## Methods

**Sample collection and sequencing**. An adult male great bustard was found dead on the 17th of January, 2022, at Hesheng Town, Ning County, Qingyang City, Gansu Province, China (35°431255′N, 107°782055′E). The local temperature was −7 to 4 °C during the day we collected the animal. The frozen individual was immediately relocated to the laboratory for dissection. Five tissues were collected for sequencing, including leg thigh muscle, brain, heart, lung, and liver. We had a previously preserved sample of a female great bustard, but its low preservation quality only allowed for short-read but not long-read sequencing. The Institutional Animal Care and Use Committee (IACUC) of Longdong University has approved the animal ethics.

Genomic DNA for ultra-long reads sequencing was isolated from the thigh muscle. Ultra-long DNA was extracted by the Phenol:Chloroform:Isoamyl alcohol (25:24:1) method from the Tris+SDS (Sodium Dodecyl Sulphate)+EDTA+NaCl lysing reagents treated[64] tissues without a purification step to ensure a sustained length of genomic DNA. Then, length >50 kb genomic DNA was size-selected in the SageHLS HMW library system (Sage Science, USA), and sequencing libraries were processed by using the Ligation sequencing 1D kit (SQK-LSK109, Oxford Nanopore Technologies, UK) according to the manufacturer's instructions. Three DNA libraries were constructed and sequenced on the PromethION (Oxford Nanopore Technologies, UK) at the Genome Center of Grandomics (Wuhan, China).

For short-read sequencing, the cetyltrimethylammonium bromide (CTAB) method was used to extract genomic DNA from thigh muscle. Genomic DNA was randomly fragmented, then an average size of 200–400 bp fragments was selected by Agencourt AMPure XP-Medium kit. The constructed library was then sequenced on the MGISEQ-T7 platform (Genome Center of Grandomics,Wuhan, China)[65] with PE 150 bp mode.

Hi-C sequencing data was used to anchor assembled contigs onto chromosomes. Muscle tissues were cut into pieces in nuclei isolation buffer supplemented with 2% formaldehyde for cross-linking. Cross-linking was stopped by adding glycine and additional vacuum infiltration. Fixed tissue was then grounded to powder and resuspended in nuclei isolation buffer to obtain a suspension of nuclei. To remove unligated DNA fragments, the purified nuclei were digested with 100 units of Dpnll and marked by incubating with biotin-14-dATP. The ligated DNA was sheared into 300–600 bp fragments, and then followed by a standard library preparation protocol[66]. Hi-C sequencing was

conducted on the MGISEQ-T7 platform (Genome Center of Grandomics,Wuhan, China) with the PE 150 bp mode.

**Genome assembly and assessment**. We used three cells of Nanopore ultra-long reads for de novo assembly with default parameters in Nextdenovo v2.5.0[67]. To correct assembly errors, we applied two rounds of polishing with Nextpolish v1.4.0[68] based on the long-read alignments. We further polished the assembly twice with short reads using Pilon v1.24[69]. Benchmarking Universal Single-Copy Orthologues (BUSCO V5.2.2)[70] with the aves_odb10 lineage ($n = 8338$) was used to assess genome assembly completeness.

**Chromosome-level assembly with Hi-C data**. We mapped the reads from the Hi-C library sequencing against the contigs with the Juicer (v1.6) pipeline[71]. The ".hic" file was generated using the 3D-DNA (v180419) pipeline[72] and the Hi-C heatmap was visualized in the Juicebox Assembly Tools[73] for manual curations. We demarcated chromosome boundaries and reverse or reorder contigs according to the Hi-C interaction heatmap in Juicebox. The MCscan function from the JCVI package[74] was used to identify synteny blocks among the great bustard, chicken and zebra finch. Default parameters were used except for 'minispan = 30' when performing jcvi.compara.synteny screen process.

**RNA-sequencing and transcriptome assembly**. Total RNA from thigh muscle, brain, heart, lung, and liver tissues were extracted using the Trizol (Invitrogen, Carlsbad, CA, USA) method. RNA-sequencing libraries were prepared and sequenced on the MGISEQ-T7 platform with the PE 150 bp mode. After trimming RNA-seq reads with Trimmomatic (v0.39, default parameters)[75], we mapped the filtered RNA-seq against the genome assembly with HISAT2 (v2.1.0)[76]. Transcripts were assembled using StringTie (v2.0)[77]. TransDecoder (v5.5.0)[78] was used to predict protein-coding regions of the assembled transcripts.

**Repeat annotation**. Avian homology repetitive elements were acquired from RepeatMasker (http://www.repeatmasker.org) database (RepeatMaskerLib.h5). EDTA (v2.0.1)[79], TRF (v4.09)[80], and RepeatModeler (v2.0.1) were used for de novo prediction of repetitive elements. Tandem repeats predicted by TRF[80] were filtered by the pyTanFinder pipiline[81]. CD-hit (v4.8.1)[82] was used to construct a non-redundant repeat library based on all de novo and known libraries. RepeatMasker (v4.1.2-p1) was used to mask repetitive elements in assembled chromosome-level genome with default parameters.

**Genome annotation**. For homolog-based methods, genome threader (v1.7.1)[83] and exonerate (v2.4.0)[84] (implemented in maker3[85]) were used to predict gene models. Chicken and zebra finch protein sequences were used as queries for homolog search. For ab initio-based methods, AUGUSTUS v3.4.0[86], GeneID v1.4[87] and SNAP[88] were used to predict gene models. The training models of AUGUSTUS were directly acquired based on BUSCO (V5.2.2) gene models. The training set for GeneID was derived from transcriptome-based evidence. The initial training set of SNAP (v2006728) was acquired from homolog proteins and genome-guided transcripts assembled with TRINITY v2.8.5[89]. Three-round training was performed in SNAP. We then used EVM v1.1.1[78] to integrate all predicted gene models by the above three methods. Finally, we used the PASA (v2.5.2)[90] pipeline to polish gene models using TRINITY genome-guided transcript assembly.

**DNA methylation**. DNA 5mC methylation was called with Nanopolish (v1.4.0)[91] by using the Hidden Markov Model. ONT ultra-long fast5 files were used as the input files. The methylation frequency was calculated as the number of reads on methylated cytosine (xmCpG) divided by the total number of reads covering each cytosine site in the reference (xmCpG+xCpG). We further calculated the mean methylation levels over 50 kb windows, and analyzed the correlations with other genomic features.

**Phylogenetic analyses**. Coding sequences of chicken (*Gallus gallus*, ASM2420605v1)[30], zebra finch (*Taeniopygia guttata*, GCA_003957565.4)[22], kori bustard (*Ardeotis kori*, ASM1339637v1)[19], Common Cuckoo (*Cuculus canorus*, ASM70932v1)[18], Rock pigeon (*Columba livia*, Cliv_2.1)[92], grey go-away-bird (*Corythaixoides concolor*, ASM1339949v1)[19] and the great bustard (this study) were used to construct phylogenetic tree and to perform comparative genomics analyses. To do so, we used OrthoFinder2[93] to identify single-copy orthologous genes for the sampled bird species. IQTREE2[94] was used to construct the maximum likelihood tree based on single-copy orthologous genes, and the best substitution model JTT + F + I + I + R4 automatically selected by ModelFinder[95] for protein alignments. For coding sequence (CDS) alignment, the best substitution model GTR + F + R3 was automatically selected by ModelFinder. We used LAST (v1066)[96] to perform whole-genome pairwise alignment with great bustard as the reference, and used MULTIZ (v11.2)[97] to generate multiple whole-genome alignments. Only one-to-one alignments were retained, and the alignments were further filtered by the TrimAl (v1.2)[98] strict model. The best substitution model GTR + F + R5 was used.

The approximate likelihood calculation method[99] in PAML-MCMCTREE (PAML v4.9j)[100] was used for divergence time estimation. We used PAML-CodeML[100] to acquire an initial branch length with the gradient and Hessian information for the ultrametric tree. Then, this file was taken as the input for MCMCTREE[100] to run 20 million steps of MCMC chains to estimate divergence time. The fossil calibration confidence interval of Neognath and Neoaves (60.2–86.8 MYA) was derived from the Palaeontologia Electronica Fossil Calibration Database[101].

**Gene family evolution**. The orthogroups identified by OrthoFinder2 was used as input to Café (v4.2.1)[102] to infer gene family expansions and contractions. The ultrametric tree topology and branch lengths from the MCMCTREE were used to infer the significance of changes in gene family size in each branch, and significant levels of expansion and contraction (*P* value) were determined at 0.05. We conducted GO (Gene Ontology) enrichment analysis for genes in the expanded families with PANTHER17.0[103].

**Positive selection**. We estimated the nonsynonymous to synonymous substitution rate ratios (ω = dN/dS) to assess selection on single-copy orthologous genes. The single-copy orthologous nucleotide sequences were generated by ParaAT 2.0[104] which generated back-translated nucleotide alignments guided by protein-coding sequences alignments to ensure the alignments were reliable and accurate. MAFFT (v7.505)[105] was used as the aligner. All gaps were automatically removed by ParaAT, and alignment lengths shorter than 99 bp were removed. We estimated ω by using the branch-site model[106,107] implemented in the CodeML program in PAML[100], using unrooted trees to fit the parameter clock = 0. The branch-site modelA estimated ω values for each site in foreground and background, and then divided sites into three categories: ω < 1 (ω0), ω = 1 (ω1), and ω > 1 (ω2). The null model of the branch-site modelA (modelA null) does not allow ω larger than 1 in all branches. The alternative hypothesis allowed ω values to be larger than one in the foreground branch (great bustard), representing the positively selected sites. *P*-values were calculated through the likelihood ratio test and then adjusted by false discovery rate (FDR) corrections[108]. Genes with ω2 higher than 1, FDR smaller than 0.01 and positively selected sites (2a and 2b) more than 5% were considered as positively selected genes (PSGs). We further performed additional two rounds of branch-site model analyses. PSGs missing in any verification round were filtered. Candidate PSGs were taken for GO enrichment analyses in PANTHER17.0[103]. Fixed Effects Likelihood (FEL)[109] was used to estimate positively selected sites, with kori bustard, great bustard and tMRCA labeled as test branches. Profile likelihood confidence intervals for each variable site were computed.

**Real-time RT-qPCR**. Total RNA used in reverse transcription–qPCR was reverse-transcribed to cDNA by using Takara® PrimeScript™ RT reagent Kit with gDNA Eraser (Perfect Real Time) (Takara, Japan). Two μg of total RNA and 1 μl gDNA Eraser with 2 μl 5 × gDNA Eraser Buffer were incubated at 42 °C for 2 min to remove genome DNA (gDNA), then 5 μl 5 × PrimeScript Buffer 2, 1 μl PrimeScript RT Enzyme Mix I and 1 μl RT Primer Mix was added to the 10 μl gDNA removed reaction mixture and incubated at 37 °C for 15 min and 85 °C for 5 s. The cDNA was used for real-time PCR detection in Applied Biosystems 7500 Real Time PCR System (ABI, USA) using TB Green Premix Ex Taq II (Tli RNaseH Plus) (Takara, Japan). The condition for Quantitative RT-PCR was 95 °C for 5 min, followed by 40 cycles of 95 °C for 10 s and 60 °C for 34 s. All reactions were run in triplicate, we used 2^-ΔΔCt equation (Formula 1) to calculate relative expression fold changes.

$$^{\Delta\Delta}Ct = (CT\ target\ gene - CT\ target\ actin\beta) - (CT\ control\ gene - CT\ control\ actin\beta) \quad (1)$$

The primer sequences used to amplify each target gene can be found in Supplementary Table 11.

**Demographic analyses**. The PSMC model was used to infer demographic history of the great bustard. We used interval parameters "4 + 30 * 2 + 4 + 6" sets in PSMC (v0.6.5), where "4 + 30 * 2 + 4 + 6" has 74 atomic intervals distributed across 33 free intervals[110] which has been used for several bird genomes[111–113]. We performed 100 times bootstraps to estimate the variance of the simulated results. The neutral mutation rate μ (mutations per base per generation) used for PSMC was calculated by the Formula 2,

$$\mu = D*g/2T \quad (2)$$

Where D and T were the estimated genome-wide nucleotide divergence and the estimated divergence time between the great bustard and kori bustard, and g was the estimated generation time. Females and males of great bustard have different maturation ages, with males usually starting to mate from 5 to 6 years of age, while females at 2 to 3 years old[26]. We assumed a generation time of 3 years. The genome-wide nucleotide divergence was calculated by nucmer and dnadiff from the MUMmer program (v4.0.0)[114,115].

**Genetic diversity**. Genetic diversity was assessed by calculating individual heterozygosity. Short reads were mapped to genome sequence by using the BWA-MEM algorithm (v0.7.17)[116]. Picard package (v2.25.0)[117] was used to mark duplicates. The Genome Analysis Toolkit (GATK v4.2.0.0)[118]. HaplotypeCaller module was used for variant calling. SNPs genotypes were generated from GATK GenotypeGVCFs module. Variant quality information Quality by depth (QD) < 2.0, Fisher Strand (FS) > 60.0, RMS Mapping Quality (MQ) < 40.0, Mapping Quality Rank sum (MQRankSum) <−12.5, and Position Rank Sum (ReadPosRankSum) <−8.0 were used to filter low-quality variants by using the VariantFiltration module. Heterozygosity was calculated by dividing the total number of heterozygous sites by the genome size covered by reads.

**Runs of homozygosity**. We generated short-reads from an additional female great bustard individual to perform ROH analysis, together with the male short reads. Short reads were mapped to OtiTar_swu by Bwa-Mem, and then Picard and GATK4 was used to mark duplicates, SNP calling and filter low quality SNPs, the same as described in above "Genetic diversity" section processes. Bcftools (v1.9)[119] was used to merge two indivaduls' VCF files and keep bi-allelic SNPs. Plink (v1.90b4)[120] was used to filter SNPs with minor allele frequency (MAF) less than 1%. The *homozyg* module in Plink was used measuring runs of homozygosity, and parameters "--homozyg-window-snp 50 --homozyg-snp 50 --homozyg-kb 100 --homozyg-density 50 --homozyg-gap 1000 --homozyg-window-missing 5 --homozyg-window-threshold 0.05 --homozyg-window-het 03" was used.

**Sex chromosome evolution**. We mapped resequencing reads from a female individual to the reference genome using BWA MEM. We used Samtools (v1.9)[121] depth to calculate sequencing coverage with default parameters. We also calculated sequencing coverage for alignments with a stringent filtering criterion, *i.e.*, only retaining reads having no more than 2 mismatches, to avoid ZW cross-mapping. We then used Bedtools (v2.29.1)[122] to calculate the mean sequencing converge over 50 kb windows. GATK was used to calculate heterozygosity, variant quality information Quality by depth (QD) < 5.0, Fisher Strand (FS) > 20.0, RMS Mapping Quality (MQ) < 50.0, Mapping Quality Rank sum (MQRankSum) <−12.5, Position Rank Sum (ReadPosRankSum) <−8.0 and "-window 30 -cluster 3" were used to filter low-quality variants by using the VariantFiltration module.

**Statistics and reproducibility**. Assumptions of normality of variance were examined by Kolmogorov–Smirnov tests performed in IBM SPSS v26. Linear regression analyses were performed by the lm function in R (v4.2.0), for the approximate *p*-values derived from lm function, further robustness tests were performed in JASP (v0.16)[123]. ANOVA and its post hoc analysis (Tukey HSD) were performed in the IBM SPSS v26. Mann–Whitney test were performed in the IBM SPSS v26. P-values derived from ANOVA and Mann-Whitney test were reported in two-sided tests, and were adjusted by false discovery rate (FDR) correction[108] with a cutoff of 0.05.

**Reporting summary**. Further information on research design is available in the Nature Portfolio Reporting Summary linked to this article.

## Data availability

The genome assembly data are available at GenBank under the accession number: JAPMTP000000000. Genomic sequencing reads and RNA-seq data are deposited in the BioProject PRJNA903785. Gene annotations and source data for Figs. 3–6 are deposited in a FigShare repository (https://doi.org/10.6084/m9.figshare.23650419)[124].

## Code availability

Custom scripts used in this study have been deposited at Github (https://doi.org/10.5281/zenodo.8127516)[125].

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

## Acknowledgements

We thank Junjie Yin (Xiamen University) for assistance in figure preparations. This study is supported by the "Special fund for youth team of the Southwest University" (SWU-XJPY202302) to LX.

## Author contributions

L.X., H.L., and X.O. conceived the project and designed the experiments. H.L., X.J and J.W. and J.S. performed genome sequencing and assembly. H.L., X.J, and Z.X. performed data analyses. X.Z., Z.H., B.L, and M.H. participated in the project design and provided samples. H.L. and L.X. wrote the manuscript and revised the manuscript. All authors read and approved the final manuscript.

## Competing interests

The authors declare no competing interests.
