## [Peer Review File · Communications Biology]

Reviewers' comments:

Reviewer #1 (Remarks to the Author):

The manuscript "Retrieving the near-complete genome of a threatened bird from wild frozen samples" by Luo et al presents a genomic assembly and subsequent downstream analyses of the great bustard. They use a specimen sample that was found dead in wintry conditions beginning of 2022, although causes of death etc are not known. Generally this is a very complete genome assembly using Nanopore sequencing complemented by HiSeq, RNAseq and short read sequencing. This is a well written manuscript covering all major aspects of a single genome contributed to the community. Because it uses Nanopore sequencing as a major technology, which is also used for epigenetic analysis I think it has a novelty aspect to it that justifies publication. My major criticism is that it is argued that this is a very good example of obtaining samples from threatened species. Well I understand the sentiment of it, however the presented work does not comprehensively investigate this aspect - as I understand the authors found the specimen by chance, and the knowledge of the conditions of the specimen is very limited. So while the genome assembly looks very reasonable, no knowledge can be really obtained and a guideline developed for cold-preserved samples. I think for this a more systematic study on degradation conditions and temperature conditions surrounding the death of the animal would be necessary. I therefore would recommend that the authors may change their story line for this otherwise fine work.

Specific comments:

Title: it is unclear what near complete means - perhaps high quality. was the sample really frozen? In the text above 0 temperatures are mentioned. Generally I find the title too vague.

l29 I find the abbreviation OTswu not very intuitive

l160 You mentioned that you used nanopolish, but did you try other methods as well?

l200 Branch-site models are really vulnerable to alignment artifacts. Did you use other aligners (such as clustalo or prank) to investigate the alignment method on the outcome of the PSG

l203 you mention that you used branch-site model - but it is unclear which one. I suspect it's BS model A, but please clarify this

l206 which one did you use at the foreground branch?

l221: you set $g=3$, but highlighting the different maturation times between the sexes. Is it hence a reasonable estimate? how do you justify 3?

l232: how did you deal with repetitive regions.

l 338: I doubt that exponentials are written like this

l67 and l 368: 500 versus 800, please clarify

I think what is missing is a runs of homozygosity analyses because heterozygosity is really low, and there is no a-priori knowledge whether the specimen is a good representative of the species (e.g. inbred individual). examples can be found for the Alpine marmot genome or the Iberian Lynx.

I think there is no statement regarding availability of scripts used in this analysis. This should be standard by now.

Fig3: what is the outlier in a and b (bottom right corner)?

Fig4a. what is on the axes?

Fig4c: x-axis unlabeled

Fig 4d: what is the unit here (says 3.47 versus 0.37)

Fig5c: dip at the end?

Reviewer #2 (Remarks to the Author):

The authors found a dead frozen individual of the great Bustard and managed to extract high quality DNA to be used for ONT sequencing and short read sequencing as well as RNA for transcriptome profiling. They assembled and annotated a high quality reference genome which they used for downstream population genetic and molecular evolution analyses. The authors reconstructed the demographic history, phylogenetic relationships and gene family evolution.

While the study is quite interesting and solid, and somewhat relevant for the conservation genomics field, I do question whether this study fits this particular journal. Maybe, for example a genome report in some other journal would be a better fit.

Comments:

- I would like to see some additional results on the ONT sequencing, such as the distribution of read lengths.

- Was any initial genome size estimation carried out? For example on the short read data using genomescope or similar?

- For the purpose of this study it would also be useful to include quality measurements of the RNA used for the transcriptomic part of the study.

-(L. 130) Explain what your manual curation involved.

- (L.166-L.187) I find the phylogenomic part in the method section very confusing as it does not spell out how and why the data is treated in a particular way after the ortholog detections. The only information consists of what substitution model was the most optimal for CDS and protein alignments. I am also not quite sure what the phylogenetic tree that is presented in the study is based on. Consider re-writing this section and state more clearly what data has been used for what.

- I am missing any mention in the methods how the synteny analysis was carried out.

- I am a bit puzzled by the use of the term dot chromosomes. This terminology comes from the Drosophila literature and is at least, as far as I know not commonly used in the avian genomics literature. Consider changing it to dot-shaped/like micro-chromosomes (if that's what it actually is).

- In terms of assembly statistics, why such a focus on contig N50 over scaffold N50? Table.S2 contains more information that could be mentioned in the assembly results section.

-(L.47) This is clearly not a major challenge for only large animals.

-(L.53 and beyond) Instead of writing of a recent death, use for example recently deceased or has recently died.

-(L.57) maybe they are rare, but there is no reason to believe that frozen avian tissue would behave any different compared to mammals. Bringing up the wholly mammoth here is not really necessary for comparisons sake as we are talking about very different time spans of tissue preservation.

-(L.59 – L.63) I am personally not at all impressed by the notion of a digital Noah's Ark, but since the authors use it in the introduction there should at least be a stated purpose of such an endeavor.

- (L.97) What do the authors actually mean with the local temperature was -7 to 4C? During what period? What was the temperature during collection?

-(L.101) State which muscle.

-(L.102) Spell out SDS and possibly cite.

-(L.111) A mention where these MGISEQ machines ran and citation.

-(L.149) Implemented in maker3. Were used to predict gene models.

-(L.154-157) I assume that the authors mean genome guided and not guild.

-(L.189) Was used as input.

-(L.214-216) Consider restructuring this section and not start the sentence with what interval parameters that you used. Also, specify that those interval parameters are for atomic time as well as free parameters in the model. There is also the necessary steps leading up to the actual PSMC analysis missing.

-(L.219) What was the resulting mutation rate?

-(L.234) We mapped.

-(L.235) calculate sequencing coverage, not converge.

-(L.238 – 239) I assume the authors used bedtools to subset the data in order to calculate coverage over windows. Consider rephrasing.

-(L.282) Just phylogenomic? Needs a new sub header.

-(L.318) Adaptive to powered flight? Needs a new sub header

-(L.338) $6.11e10^{-9}$, what does this mean?

-(L.343) Be careful with the interpretation of PSMC results in recent times. The method is not really capable of yielding accurate estimates in recent times.

Response to Editors

(1) Reframe the manuscript to focus on the great bustard genome assembly and its evolutionary history, rather than claim that the study could be used as a framework to obtain frozen samples from a threatened species (as noted by Reviewer #1).

R: As we respond to Reviewer #1 below, we have now extensively revised the manuscripts, changed the title to "A high-quality genome of great bustard recovered from a recently deceased individual", removed any sentence hinting that our study serves as a guideline for cold-preserved samples, and added a new paragraph (line 251-257) in the discussion.

(2) Please include the run of homozygosity analysis (per Reviewer #1), and the additional metrics (RNA quality, genome size estimation, etc) outlined by Reviewer #2.

R: We have added the run of homozygosity analysis (Fig. 5c) and metrics for RNA quality (Table S9), genome size estimation (Fig. S3) and ONT read distribution (Fig. S2).

(3) Expand on the Methods, as noted by both reviewers. Please note that this section does not have a word limit and should contain sufficient detail such that readers could feasibly repeat these analyses themselves.

R: we have now extensively revised the Methods, added more details or missing content pointed out by the reviewers.

(4) From an editorial perspective, we would ask that you expand on the transcriptomic analyses and comment on any tissue-specific gene expression patterns, while also including secondary validation of any key tissue-specific genes (e.g. qRT-PCR).

R: We have performed additional analyses to screen for genes that are under positive selection and at the same time show tissue-specific expression patterns. Using this approach, we identified two genes, *MYOD1* and *MYF5*, that were positively selected and have muscle-specific high expression. We discussed the possible role of *MYOD1* and *MYF5* in the adaptation evolution of great bustard, and have also validated the expression patterns of *MYOD1* and *MYF5* using qRT-PCR. The results are reflected in Fig. 4 and in the Results (line 189-196) and Discussion (line 254-271).

Response to Reviewers

Reviewer #1 (Remarks to the Author):

The manuscript "Retrieving the near-complete genome of a threatened bird from wild frozen samples" by Luo et al presents a genomic assembly and subsequent downstream analyses of the great bustard. They use a specimen sample that was found dead in wintry conditions beginning of 2022, although causes of death etc are not known. Generally this is a very complete genome assembly using Nanopore sequencing complemented by HiSeq, RNAseq and short read sequencing. This is a well written manuscript covering all major aspects of a single genome contributed to the community. Because it uses Nanopore sequencing as a major technology, which is also used for epigenetic analysis I think it has a novelty aspect to it that

justifies publication. My major criticism is that it is argued that this is a very good example of obtaining samples from threatened species. Well I understand the sentiment of it, however the presented work does not comprehensively investigate this aspect - as I understand the authors found the specimen by chance, and the knowledge of the conditions of the specimen is very limited. So while the genome assembly looks very reasonable, no knowledge can be really obtained and a guideline developed for cold-preserved samples. I think for this a more systematic study on degradation conditions and temperature conditions surrounding the death of the animal would be necessary. I therefore would recommend that the authors may change their story line for this otherwise fine work.

R: We appreciate the reviewer's positive remarks on our manuscript, and we agree that we need to change the storyline of this work by focusing more on the great bustard. We have now extensively revised the manuscripts, changed the title to "A high-quality genome of great bustard recovered from a recently deceased individual", removed any sentence hinting that our study serves as a guideline for cold-preserved samples, and added in the discussion that "We used the samples collected from a recently died great bustard that had been surprisingly well preserved. Given that we had sparse information about the cause and date of the death or other conditions that may impact the degradation of DNA and RNA of the dead animal, our study provides limited knowledge or guideline for conducting a similar study when a dead animal is newly spotted. Nevertheless, the great bustard samples we obtained provided valuable resources for conservation genomics. We call for swift action to translocate dead animals to laboratories that can be fully utilized for genomic and other biological analyses."

Specific comments:

Title: it is unclear what near complete means - perhaps high quality. Was the sample really frozen? In the text above 0 temperatures are mentioned. Generally I find the title too vague.

R: We have now changed the title to "A high-quality genome of great bustard recovered from a recently deceased individual". Hope now it's clearer.

I29 I find the abbreviation OTswu not very intuitive

R: We have changed the assembly name to OtiTar_swu, following the nomenclature rule proposed by the VGP (Vertebrate Genome Project).

I160 You mentioned that you used nanopore, but did you try other methods as well?

R: Unfortunately we didn't. We had a good experience working with nanopore in another bird species (Huang et al. 2023. *PNAS*), so we decided to continue to use nanopore in this study.

I200 Branch-site models are really vulnerable to alignment artifacts. Did you use other aligners (such as Clustal or Prank) to investigate the alignment method on the outcome of the PSG?

R: Thanks for pointing out this concern which we agree is very important in identifying PSGs. In this study, we used the ParAT pipeline which involves Epa2nal (modified from PAL2NAL) to back-translate the nucleotide alignments guided by amino acid alignments, which has been suggested to be more reliable and accurate than direct nucleotide alignment (Zhang et al. 2012. *Biochem Biophys Res Commun*; Suyama et al. 2006. *Nucleic Acids Research*). Mismatches

(mainly caused by insert/delete frameshift) or incorrectly aligned sites have mostly been filtered out by ParaAT. We have also manually inspected the alignments for the detected PSGs.

I203 you mention that you used branch-site model - but it is unclear which one. I suspect it's BS modelA, but please clarify this

R: Thanks for pointing this out. The branch-site modelA was used as the alternative model and modelA null was used as the null model in our study. We have clarified this in our revised MS at line 357 and 359.

I206 which one did you use at the foreground branch?

R: The great bustard we set as the foreground branch. We have added a note in our revised MS at line 399.

I221: you set $g=3$, but highlighting the different maturation times between the sexes. Is it hence a reasonable estimate? how do you justify 3?

R: The reason why we used $g=3$ was based on the female maturation time which is 2 to 3 years. Unfortunately, there was sparse data about the female maturation time. According to Dr. Xiaobin Ou (a co-author of this study) who has been involved in great bustard conservation work, the maturation age was close to 3 years.

I232: how did you deal with repetitive regions.

R: We should have treated the repetitive regions differently. A part of the reason we did not was that bird genomes in general have a low repeat content (~10%, according to Zhang *et al.* 2014 Science) and most transposable elements are probably not recently propagated. For the young transposable elements or satellite DNA, the short-read alignments likely either have low alignment scores (due to multiple hits) or low coverage - so variants called from those regions are more likely to have been filtered out (we have not evaluated this by ourselves though). For those reasons, we believe the repetitive regions have a limited impact on variant calling for a bird genome.

I338: I doubt that exponential are written like this

R: We apologize for this typo, we have corrected it in the revision MS.

I67 and I 368: 500 versus 800, please clarify

R: In line 67, 500 means the numbers of sequenced birds, but in line 368, 800 represents available assemblies. Because many birds had more than one assembly version, the numbers of assemblies are larger than that of sequenced birds. We have now changed 800 to 500 for consistency.

I think what is missing is a runs of homozygosity analyses because heterozygosity is really low, and there is no a-priori knowledge whether the specimen is a good representative of the species (e.g. inbred individual). examples can be found for the Alpine marmot genome or the Iberian Lynx.

R: Yes, the individual we used might not be a good representative. Fortunately, we also had a sample from a female as well as short-read sequencing data. During the revision, we analyzed the runs of homozygosity for two great bustard individuals. The result showed the great bustard had a moderate level of F_{ROH} from 0.076 to 0.088. The average lengths of ROH were about 300 Kb. We have added the following sentences in the main text:

“To evaluate whether inbreeding may have caused the reduced heterozygosity, we calculated the lengths of ROH (run of heterozygosity) using short-read sequencing data from a male (the one used for genome assembly) and a female individual. The average lengths of ROH were 0.27 and 0.36 Mb, covering 7.6% and 8.8% of the genome (F_{ROH}) for the male and female individual, respectively. The F_{ROH} is smaller than that in kakapo (>15%) or California condor (>20%), but larger than that in Tibetan chicken (4 - 6%) and many other non-threaten birds.”

I think there is no statement regarding availability of scripts used in this analysis. This should be standard by now.

R: Thanks for the suggestion about the code availability. We deposit our analysis scripts at GitHub, and the website can be found in the **Data Accessibility and Benefit-Sharing** section.

Fig3: what is the outlier in a and b (bottom right corner)?

R: it is chromosome 31 which had the highest GC content.

Fig4a. what is on the axes?

R: The X axis represents count numbers, and the Y axis represents higher FDR to lower FDR. We have now added the titles beneath the axes. Note this figure has been moved to supplementary figures.

Fig4c: x-axis unlabeled

R: We have now added the label for the x axis.

Fig 4d: what is the unit here (says 3.47 versus 0.37)

R: The unit is the percentage of heterozygous sites. We have now added the label.

Fig5c: dip at the end?

R: yes, the last windows contain very few aligned sites (too repetitive). Now we have removed the windows that contain more than 40% unaligned sites.

Reviewer #2 (Remarks to the Author):

The authors found a dead frozen individual of the great Bustard and managed to extract high quality DNA to be used for ONT sequencing and short read sequencing as well as RNA for transcriptome profiling. They assembled and annotated a high quality reference genome which they used for downstream population genetic and molecular evolution analyses. The authors reconstructed the demographic history, phylogenetic relationships and gene family evolution.

While the study is quite interesting and solid, and somewhat relevant for the conservation genomics field, I do question whether this study fits this particular journal. Maybe, for example a genome report in some other journal would be a better fit.

Comments:

- I would like to see some additional results on the ONT sequencing, such as the distribution of read lengths.

R: Thanks for the suggestion. We have added the read length distribution in a supplementary figure (also attached below).

- Was any initial genome size estimation carried out? For example on the short read data using genomescope or similar?

R: Yes, we did estimate the genome size using genomeScope (line 99) but did not present the data in the original manuscript. In the revised manuscript, we added the genomeScope estimation in a supplementary figure (also attached below).

- For the purpose of this study it would also be useful to include quality measurements of the RNA used for the transcriptomic part of the study.

R: Yes we agree, and we have added RNA quality information in Table S9 (also attached below):

Table S9 RNA and RNA sequencing bases quality of 5 tissues

Tissue	A260/A280	A260/230	RIN	C (ng/ μ l)	Q20(B/A)	Q30(B/A)
Leg thigh muscle	1.99	1.27	5.7	451.0	97.62%-97.66%	93.57%-93.63%
Brain	2.03	2.03	8.2	1896.0	97.51%-97.55%	93.40%-93.46%
Heart	1.98	1.65	5.1	670.0	97.46%-97.52%	93.56%-93.65%
Lung	1.95	1.39	8.3	828.0	97.54%-97.58%	93.36%-93.43%
Liver	1.98	1.49	6.5	1905.0	97.31%-97.40%	93.28%-93.43%

C: Concentration. B/A: Bases quality before and after filtered by Fastp program.

-(L. 130) Explain what your manual curation involved.

R: We demarcated chromosome boundaries and reverse or reorder contigs according to the Hi-C heatmap visualized in Juicebox. We have added such details in the revised manuscript at line 318

- (L.166-L.187) I find the phylogenomic part in the method section very confusing as it does not spell out how and why the data is treated in a particular way after the ortholog detections. The only information consists of what substitution model was the most optimal for CDS and protein alignments. I am also not quite sure what the phylogenetic tree that is presented in the study is based on. Consider re-writing this section and state more clearly what data has been used for what.

R: Many thanks for this constructive comment. We have revised this section in the revised MS at line 144-150. We hope the writing is now clearer.

- I am missing any mention in the methods how the synteny analysis was carried out.

A: We used the MCscan method to analyze the synteny. In the revised ms, we have added the description of the MCscan workflow in the Methods: “The MCscan function from the JCVI package was used to identify synteny blocks among the great bustard, chicken and zebra finch. Default parameters were used except for ‘minispan=30’ when performing `jcvi.compara.synteny` screen process.”

- I am a bit puzzled by the use of the term dot chromosomes. This terminology comes from the Drosophila literature and is at least, as far as I know not commonly used in the avian genomics literature. Consider changing it to dot-shaped/like micro-chromosomes (if that's what it actually is).

R: Yes, we were analyzing the dot-like microchromosomes. We borrowed the term from a recently published chicken genome (Huang et al. 2023 [10.1073/pnas.2216641120](https://doi.org/10.1073/pnas.2216641120)). The “dot chromosomes” show a high level of chromosome homology between chicken and great bustard. In order to be consistent with the descriptions of the dot-like microchromosomes in Huang *et al.* (2023), we also used the term “dot chromosome” in this study.

- In terms of assembly statistics, why such a focus on contig N50 over scaffold N50? Table.S2 contains more information that could be mentioned in the assembly results section.

R: Yes, we missed the descriptions of scaffold N50 in the original ms. Now we have included the scaffold N50 result in Table S3 and in the main text.

-(L.47) This is clearly not a major challenge for only large animals.

R: Yes, we agree with the reviewer, and have removed the word “large”.

-(L.53 and beyond) Instead of writing of a recent death, use for example recently deceased or has recently died.

R: Many thanks for this advice, we have revised the writing throughout the manuscript.

-(L.57) maybe they are rare, but there is no reason to believe that frozen avian tissue would behave any different compared to mammals. Bringing up the wholly mammoth here is not really

necessary for comparisons sake as we are talking about very different time spans of tissue preservation.

R: Yes, we agree, and have removed the mention of mammoth in the revised manuscript.

-(L.59 – L.63) I am personally not at all impressed by the notion of a digital Noah's Ark, but since the authors use it in the introduction there should at least be a stated purpose of such an endeavor.

R: Many thanks for this constructive comment, we added "This is critical to preserve complete genomic information for endangered species, an endeavor proposed by some initiatives such as the digital Noah's Ark".

-(L.97) What do the authors actually mean with the local temperature was -7 to 4C? During what period? What was the temperature during collection?

R: We re-wrote the sentence and hope it is clearer now: "The local temperature was -7 to 4 °C during the day we collected the animal"

-(L.101) State which muscle.

R: Leg thigh muscle was used, we had added it in line 93.

-(L.102) Spell out SDS and possibly cite.

R: We have now spelled out SDS (Sodium Dodecyl Sulphate) in the revised manuscript in the line 287.

-(L.111) A mention where theses MGISEQ machines ran and citation.

R: We have added more information about the MGISEQ sequencing the in the revised manuscript in line 298.

-(L.149) Implemented in maker3. Were used to predict gene models.

R: We have corrected it.

-(L.154-157) I assume that the authors men genome guided and not guild.

R: yes, that's right, we have now corrected it.

-(L.189) Was used as input.

R: We have corrected it.

-(L.214-216) Consider restructuring this section and not start the sentence with what interval parameters that you used. Also, specify that those interval parameters are for atomic time as well as free parameters in the model. There is also the necessary steps leading up to the actual

R: Many thanks for these constructive suggestions. We have rewritten the first sentence as suggested by the reviewer and specified that "4 + 25 * 2 + 4 + 6" has 64 atomic intervals distributed across 28 free intervals.

-(L.219) What was the resulting mutation rate?

R: that's 6.11×10^{-9} mutations per base per generation

-(L.338) 6.11×10^{-9} , what does this mean?

R: The mutation rate was 6.11×10^{-9} , apologize for this typo error.

-(L.234) We mapped.

R: corrected.

-(L.235) calculate sequencing coverage, not converge.

R: We have corrected it.

-(L.238 – 239) I assume the authors used bedtools to subset the data in order to calculate coverage over windows. Consider rephrasing.

R: Yes, bedtools was used only to calculate coverage over windows. We now write “We used Samtools (v1.9) depth to calculate sequencing coverage”, and “We then used Bedtools (v2.29.1) to calculate the mean sequencing coverage over 50 kb windows”

-(L.282) Just phylogenomic? Needs a new sub header.

R: We used a new subheader “Phylogenomic analysis resolve the position of Otidiformes”

-(L.318) Adaptive to powered flight? Needs a new sub header

R: We used a new subheader in the revised ms: “Identifying genes involved in Adaption to powered flight”

-(L.343) Be careful with the interpretation of PSMC results in recent times. The method is not really capable of yielding accurate estimates in recent times.

R: Thanks for pointing this out. We now added in the main text: “The low resolution of PSMC model in the recent generations, however, warrants future population-based estimation to understand the impact of recent climate changes and human activities on great bustard population dynamics”.

Reference:

Huang *et al.* (2023) Evolutionary analysis of a complete chicken genome. Proceedings of the National Academy of Sciences of the United States of America, 120(8), e2216641120.

Zhang *et al.* (2012) ParaAT: a parallel tool for constructing multiple protein-coding DNA alignments. *Biochem Biophys Res Commun.* 2012 Mar 23;419(4):779-81.

Suyama *et al.* (2006) robust conversion of protein sequence alignments into the corresponding codon alignments, *Nucleic Acids Research*, Volume 34, Issue suppl_2, 1 July 2006, Pages W609–W612.

Zhang *et al.* (2014) Comparative genomics reveals insights into avian genome evolution and adaptation. *Science* 346,1311-1320(2014).

REVIEWERS' COMMENTS:

Reviewer #1 (Remarks to the Author):

Thank you very much for the revised manuscript on the Nanopore genome assembly from the a great bustard individual. I think my raised concerns have been addressed adequately. I am a bit surprised that a second sample "suddenly" occurred in the analysis, I think this is not clearly backed up in the methods section (where does the sample come from, how was it sequenced, etc...). Also in the discussion you raise the point that female DNA would help to assemble the W - well you have this now.

Reviewer #2 (Remarks to the Author):

The authors have done a great job modifying the manuscript according to reviewer and editor comments. I am satisfied with the manuscript and think it is appropriate for publication.

Just one small note:

L.427-428: Is it 4+30 or 4+25?

Reviewer #1 (Remarks to the Author):

Thank you very much for the revised manuscript on the Nanopore genome assembly from the great bustard individual. I think my raised concerns have been addressed adequately. I am a bit surprised that a second sample "suddenly" occurred in the analysis, I think this is not clearly backed up in the methods section (where does the sample come from, how was it sequenced, etc...). Also in the discussion you raise the point that female DNA would help to assemble the W - well you have this now.

R: we had a previously stored sample that was not well preserved. However, we were able to extract DNA for short-read sequencing but not for long-read sequencing. We now explained this situation in the Methods "We had a previously preserved sample of a female great bustard, but its low preservation quality only allowed for short-read but not long-read sequencing".

Reviewer #2 (Remarks to the Author):

The authors have done a great job modifying the manuscript according to reviewer and editor comments. I am satisfied with the manuscript and think it is appropriate for publication.

Just one small note:

L.427-428: Is it 4+30 or 4+25?

R: 4+30 was used. We have revised the text.